# Phosphorous- and Boron-Doped Graphene-Based Nanomaterials for Energy-Related Applications

**DOI:** 10.3390/ma16031155

**Published:** 2023-01-29

**Authors:** Manpreet Kaur Ubhi, Manpreet Kaur, Jaspreet Kaur Grewal, Virender K. Sharma

**Affiliations:** 1Department of Chemistry, Punjab Agricultural University, Ludhiana 141004, India; 2Program for the Environment and Sustainability, Department of Environmental and Occupational Health, School of Public Health, Texas A&M University, 112 Adriance Road, College Station, TX 77843, USA

**Keywords:** doped graphene, synthetic strategies, energy devices, fuel cells, solar cells

## Abstract

Doping is a great strategy for tuning the characteristics of graphene-based nanomaterials. Phosphorous has a higher electronegativity as compared to carbon, whereas boron can induce p-type conductivity in graphene. This review provides insight into the different synthesis routes of phosphorous- and boron-doped graphene along with their applications in supercapacitors, lithium- ions batteries, and cells such as solar and fuel cells. The two major approaches for the synthesis, viz. direct and post-treatment methods, are discussed in detail. The former synthetic strategies include ball milling and chemical vapor discharge approaches, whereas self-assembly, thermal annealing, arc-discharge, wet chemical, and electrochemical erosion are representative post-treatment methods. The latter techniques keep the original graphene structure via more surface doping than substitutional doping. As a result, it is possible to preserve the features of the graphene while offering a straightforward handling technique that is more stable and controllable than direct techniques. This review also explains the latest progress in the prospective uses of graphene doped with phosphorous and boron for electronic devices, i.e., fuel and solar cells, supercapacitors, and batteries. Their novel energy-related applications will continue to be a promising area of study.

## 1. Introduction

In the last few decades, rapid industrialisation has resulted in a high demand for energy, which is coupled with the depletion of natural resources [1]. The scientific community has focused their research towards developing new materials for various catalytic applications to store energy, which are safe and eco-friendly [2,3,4,5,6,7,8,9]. Materials based on carbon, i.e., graphite, fullerene, carbon nanotubes, and graphene, are an important choice due to their non-toxic nature [10,11,12,13]. This has resulted in many applications such as engineered graphene-based membranes in desalination and water purification [14,15,16,17]. The number of products containing these nanomaterials keep on increasing because of their unique mechanical and electrical features coupled with thermal robustness [18,19,20,21]. Graphene can be chemically modified to tune its structural, electrical, and optical properties for varied applications like energy storage, generation and conversion, superconductivity, sensing, and photocatalysis [22,23,24].

The rapid progress of research on doped graphene analogues can be seen in the high number of publications (Figure 1) (record gained on Science Direct using the doped graphene keywords till July 2021). In the last decade, research on modifying the features of graphene-based nanomaterials via chemical functionalisation has shown their potential applications in various fields.

Graphene is a monolayer of graphite with a carbon–carbon bond distance of 1.42 Å and sp^2^ bound atoms of carbon organise in a honeycomb-like hexagonal framework [25,26,27]. Three ‘p’ and one ‘s’ orbital are present on each carbon atom in the graphitic lattice [28,29,30]. The two p-orbitals (p_x_, p_y_) and the s-orbital, out of the three p-orbitals, are used to produce covalent bonds with nearby carbon atoms, whilst the p_z_ orbital is employed to produce a filled π bonding molecular orbital and a vacant π* anti-bonding molecular orbital [26,28]. Due to its excellent features, such as high thermal conductivity (5300 Wm^−1^ K^−1^), low resistivity (125 m^−1^), high tensile strength (130 GPa), high electron mobility (200,000 cm^2^ V^−1^ s^−1^), and enormous specific surface area (2600 m^2^ g^−1^), graphene is also known as the “super material” of the twenty-first century [29,30,31,32,33,34,35,36]. Graphene can be transformed into graphite by stacking the graphene layers. Graphite is a 3D nanomaterial comprising of two to ten piled graphene sheets of lamellar nano dimensions [29,31]. A multilayer graphene has interesting properties, which are dependent on the number and stacking of the layers. Graphene also possesses ferromagnetism when two graphene layers are rotated by an angle [32]. The velocity of the electron declines at small twist angles and becomes zero at a certain angle, called the magic angle. This phenomenon enhances the electron’s interaction [32,33].

Upon graphite oxidation using strong acids, viz. phosphoric acid (H_2_PO_4_), sulfuric acid (H_2_SO_4_), and oxidizing agents such as permanganate (${\mathrm{MnO}}_{4}^{2-}$), numerous oxygen-bearing moieties, i.e., ketones, carboxyl, epoxides, and hydroxyl, are obtained on the surface of the graphite and the oxidised form is known as graphene oxide (GO) [37,38]. GO can be chemically modified to produce 2D channels with controlled porosity for purifying water [39,40]. Reduced graphene oxide (rGO) is the GO form that is processed using thermal, chemical, and other approaches in order to reduce the oxygen content.

The inclusion of heteroatoms with dissimilar electronegativity and size as compared to carbon atoms breaks the electroneutrality of the graphene [41,42,43,44,45,46,47] and changes the graphene nanosheet’s voltage point at Dirac point from gate voltage having a value of zero to a negative or positive side [48]. Thus, the insertion of dopants (e.g., boron (B), nitrogen (N), phosphorous (P), halogens and sulfur (S)) into the graphene sheets considerably alters their properties such as charge transport, electron mobility, Fermi level, spin density, thermal and mechanical stability, etc., and widens its applications towards sensors, solar cells, oxygen reduction reactions, lithium-ion batteries, and supercapacitors, etc. [43,49].

Heteroatoms can be doped into a graphene lattice using two approaches: (i) surface transfer, and (ii) substitutional doping. The latter approach is more effective because it entails the replacement of carbon atoms by heteroatoms [41,42,50,51,52,53,54]. In surface transfer doping, an interchange of electrons occurs between the surface-adsorbed dopant and graphene [49,55].

Tailoring the properties of graphene using nitrogen [56], halogens [57], and sulfur doping [58,59] and its applications in energy storage devices and sensors have been comprehensively reviewed. The nitrogen element, adjacent to the carbon, has almost similar atomic radii (0.70 Å) as carbon (0.77 Å), but possesses higher electronegativity (3.04) than C (2.55), which makes it easier to introduce nitrogen into the carbon lattice through substitution doping. Similar to nitrogen, the substitution of halogen atoms, viz. fluorine, chlorine, and bromine, on graphene sheets also alters the electronic properties of the host material, providing a promising way to change the transport and electronic features of the graphene for electrochemical applications. Density functional theory (DFT) states that sulphur atoms can take four different paths in doped graphene: surface adsorption, carbon atoms substitution at the graphene edge, production of oxides of S/S, and generation of sulphur clusters to create rings connecting the two layers of graphene. From an energy standpoint, sulphur is more likely to replace the carbon atoms in the serrated edge of the graphene to achieve doping. [58]. Phosphorous and boron are less electronegative than nitrogen and sulfur. Phosphorous, the “5A” group element, has a larger atomic radius as compared to nitrogen and has a higher electron donating ability [60]. Therefore, it could generate more reactive sites for the oxygen reduction reaction to improve electrocatalytic activity. This is described in a research article on the use of phosphorous-doped graphene materials as electrochemical sensors [61]. Boron represents another alternative dopant of the group “3A” that could cause p-type conductivity. The extensive use of boron-doped graphene as electrodes is also reported by Hasani and Kim [62].

The bigger size of phosphorous atoms having less electronegativity (2.19) as compared to carbon atoms (2.55), makes the C–P bond positive to some extent [63]. Phosphorus-doped graphene nanosheets (P-GNS) have a greater surface area and better separation because the length of the P–C bond (1.76 Å) is longer than the C–C bond (1.42 Å) (Figure 2a) [64,65,66]. Although boron is located in the p-block as carbon (C), it has one less valence electron [67]. Boron has an electronegativity of 2.04. It is suitable for graphene doping due to its similar size as carbon [68,69,70]. The planar configuration of carbon atoms in graphene is reserved after doping as the boron atom also forms sp^2^ hybridization in the carbon unit cell. Boron-doped graphene nanosheets (B-GNS) have different bonding configurations: (1) in-plane BC_3_ type of bonding, [71] (2) boronic acid (CBO_2_), and (3) borinic ester (C_2_BO) moieties (Figure 2b) [41,49].

The doping of graphene with heteroatoms offers redox active sites, which enhance the capacity of the probes and electrodes. Previously, a review article covering the detailed discussion on environmental application of phosphorous- and boron-doped graphene nanosheets was published [72]. In addition, a review on the boron doping of different forms of graphene (nanoribbons, graphene quantum dots) and its energy and catalytic applications was published [73]. Another review focused on boron, phosphorous, and sulphur doping of graphene was reported with a detailed discussion on its environmental applications [14]. The present review provides a detailed description of different synthesis routes of phosphorous- and boron-doped graphene, along with their respective pros and cons, and their applications in sensors, supercapacitors, lithium-ion batteries, and cells such as solar and fuel cells. Their synthesis and applications are discussed in detail in the next two sections.

## 2. Synthesis Methodologies

### 2.1. Synthesis of Phosphorous-Doped Graphene Nanosheets (P-GNS)

P-GNS can be synthesised either by direct or post-treatment methods. In direct methods, a carbon source such as benzene, toluene, alginate, and glucose, etc., is reacted with a phosphorous source, viz. phosphorous trichloride, triphenylphosphine, phosphoric acid, or phosphorous pentaoxide, to yield P-GNS. In the latter methods, previously synthesised GO is reacted with a phosphorous source such as tri-n-octylphosphine, tetradecylphosphonic, tri-n-butylphosphine acid, phosphoric acid, triphenylphosphine, polyphosphoric acid, or triphenylphosphine, to produce P-GNS. The synthesis methods, sources, and the applications of P-GNS are summarized in Table 1.

#### 2.1.1. Direct Methods

Direct pyrolysis of phosphorous trichloride (PCl_3_) and benzene (C_6_H_6_) in the temperature range of 800–1050 °C in a chamber was used to synthesise P-GNS (Figure 3) [74]. The graphitic lattice became more organised at high temperatures. The proposed chemical reaction is as below:(1)$$C_{6}H_{6}+{\;\mathrm{PCl}}_{3}\;\to {\;\mathrm{PC}}_{x}+\;\mathrm{HCl}\;+{\;H}_{2}/{\mathrm{Cl}}_{2}$$
where x=1, 2, 3, 4, 5.

Both free, stable P_4_ molecules and substitutional doping within the graphitic matrix can cause phosphorous atoms to interact with the graphitic lattice below the material’s surface. Measurements using an X-ray photoelectron spectroscopy (XPS) technique showed that phosphorous was bonded to carbon with two bonding configurations, P–C or P–P. However, the P–O bonds were also observed near the material’s surface. At a low phosphorous content, phosphorous existed in two different bonding configurations where the P–O bonds were observed at high phosphorus contents, and P–C species were difficult to detect because their signals overlapped.

Liu et al. [75] synthesised P-GNS using toluene as a carbon-containing source and triphenylphosphine as a phosphorous precursor at 1000 °C for 30 min at 600 mL min^−1^ argon flow rate. The BET surface areas of the P-GNS and the non-phosphorous material were similar.

P-GNS was synthesised by the pyrolysis of phosphoric acid as the phosphorous-containing source and alginate as the carbon-containing source at 900 °C under inert atmosphere [76]. Graphene doping with phosphorous using phosphate (${\mathrm{PO}}_{4}^{3-}$) ions was a two-step process, where firstly the reduction of ${\mathrm{PO}}_{4}^{3-}$ took place to yield phosphorous in the native form. Secondly, the subsequent reaction took place for the doping process. The substitution of carbon atoms with phosphorous in the GNS was confirmed using XPS analysis.

P-GNS having tunable porosity were fabricated using a multifunctional templating approach. An inexpensive phosphorous pentaoxide (P_2_O_5_) was used as the phosphorous precursor and glucose was used as the carbon source. The phosphorous source displayed a temperature-dependent transformation as shown in the following reaction (Figure 1) [77]:

The amount of phosphorous varied from 1.0 to 6.0 g for 2.0 g of glucose. With an increase in phosphorous content, the mesoporosity was observed to rise. The Raman spectra revealed that the defect ratio of the D to G band, i.e., I_D_/I_G_, was 1.25, which was greater than the undoped graphene (0.92). The G band is related to the C–C vibrational mode, while the D band, the defect activated band, is related to sp^2^ hybridized carbon nanomaterials. The I_D_/I_G_ ratio gives indication about the defect densities. The amount of phosphorous in P-GNS was 1.78 at.%. The insertion of phosphorous induced structural disorders into the graphitic lattice. Due to phosphorous’ excellent electron-donating qualities, this finally improved the electron delocalization. As a result, the conductivity and charge transfer were improved.

#### 2.1.2. Post-Treatment Methods

##### Thermal Decomposition

The thermal decomposition method involves chemical decomposition of a substance in the presence of heat. The reaction is endothermic in nature as heat is required to break the covalent bonds. P-GNS can be prepared by this approach using phosphoric acid, triphenylphosphine, tri-n-octylphosphine, tetradecylphosphonic, and tri-n-butylphosphine acid as doping agents. Yuan et al. used phosphoric acid to dope reduced GO. During annealing, the oxy groups were removed with the release of CO and CO_2_. The two characteristic Raman bands of reduced GO were observed at 1350 and 1590 cm^−1^, respectively. The D band displayed a red shift in P-GNS and there was a decrease in intensity of I_D_/I_G_. The XPS spectra of P-GNS displayed peaks at 134.4 eV and 133.5 eV due to P–O and P–C bonding, respectively [78]. Various topological defects and vacancies were introduced during the doping process, which increased the mesoporous nature of the materials and helped to enhance the movement and uptake of ions and electrolytes for enhancing the oxygen reduction reaction (ORR) activity.

Song and coworkers studied the formation of P-GNS using the thermal decomposition method of three different phosphorous-containing surfactants (tri-n-octylphosphine, tetradecylphosphonic, tri-n-butylphosphine acid) at two different temperatures, 800 and 900 °C. The mixed surfactant systems led to more efficient doping as compared to a single surfactant due to an increase in d-spacing (i.e., observed in the XRD measurements), which resulted in more penetration of the phosphorous atoms into this spacing [79].

Non-volatile and thermally stable 1-butyl-3-methylimidazolium hexafluorophosphate was used by Li et al. for in situ phosphorous doping of graphene nanosheets. The amount of phosphorous in P-GNS was 1.16 at.%. During the annealing process, ${\mathrm{PF}}_{6}^{-}$ reacted with the oxygeneous moieties present on the surface of GO to interact with the carbon lattice, resulting in simultaneous GO reduction and phosphorous doping. The XRD analysis suggested that the P-GNS diffraction peaks were weaker and broader, suggesting that P-doping of graphene leads to increased defects in the graphitic framework because of the significant atomic radii difference between phosphorous and carbon. This led to the observed decrease in crystallinity of the P-GNS [80].

MacIntosh et al. synthesised P-GNS from GO using triphenylphosphine as a phosphorous source at 1000 °C with or without the presence of H_2_ under argon atmosphere. The fraction of graphitic and elemental phosphorous in P-GNS prepared in the absence of H_2_ was higher than the same synthesised in the presence of H_2_. The sample was purified and any remaining high-oxidation state phosphorous species in P-GNS was eliminated by the reductive conditions formed by H_2_ at 1000 °C, as evidenced by the atomic percentage of P being 79% vs. 69% of the entire signal. The XPS studies showed different peaks at 129.6 eV, 131.0 eV, and 132.5 eV, attributed to elemental phosphorous, graphitic phosphorous, and oxidized graphitic phosphorous, respectively [81].

##### Electrochemical Erosion Methods

The electrochemical erosion method is the most effective way to create doped GNS in the presence of direct current and graphitic electrodes. Thirmual et al. prepared P-GNS using H_3_PO_4_ as an electrolyte and phosphorous source. Anhydrous H_3_PO_4_ phosphoric acid facilitated the exfoliation of the graphite layers. Upon dilution, H_3_PO_4_ dissociated into H^+^ and PO_4_^3−^ ions, which passed through the graphite layers and cleaved to form a P-GNS. The HR-TEM micrographs revealed wrinkled and folded graphene sheets. The XPS analysis of P-GNS showed 0.68 at.% phosphorus, confirming the P-doping in GNS. Moreover, the electrochemical activity of P-GNS was enhanced as compared to GNS [82].

### 2.2. Synthesis of Boron-Doped Graphene Nanosheets (B-GNS)

Different boron precursors such as triisopropyl borate, triethylborane, phenylboronic acid, diborane, boron powder, boron oxide, benzene-1, 4-diboronic acid, boric acid, and orthoboric acid can be used for the synthesis of B-GNS. Different synthesis methods along with the various sources used and the applications are summarized in Table 2. Similar to P-GNS, there are direct and post-treatment approaches for the B-GNS synthesis.

#### 2.2.1. Direct Methods

##### Chemical Vapour Deposition (CVD) Method

In the CVD methodology, the growth substrate is reacted with volatile compounds leading to the formation of the desired materials under inert conditions [98]. High-quality solid materials with excellent performance in semiconductor industries have been developed. Volatile materials are removed from the reaction chamber using the gas flow method. Cui et al. used this approach to synthesise B-GNS, where methane and ethanol served as the major source of carbon and decomposed into hydrocarbon/carbon radicals and water [84]. Trimethylborate, B(OCH_3_)_3_, acted as a boron-containing source that decomposed at a high temperature and reacted with the atoms of carbon to make B–C bonding configurations as follows:(2)$$2B\;{\left({\mathrm{OCH}}_{3}\right)}_{3}\to B_{2}O_{3}+6C\;+6H_{2}+3H_{2}O$$
(3)$${\mathrm{xB}}_{2}O_{3}+\left(2+3x\right)\;C\;\to \;2B_{x}C\;+3\mathrm{xCO}$$

The DFT calculations showed that the incorporated elemental form of boron accelerated the sp^3^ to sp^2^ conversion of the C–C bond. Cattelan et al. also synthesised B-GNS using diborane, boron-containing source and methane, and carbon-containing precursor, using polycrystalline copper foil as a growth substrate [85]. There was 1.5 at.% of boron in the material. The boron concentration was 2.5 at.% when unsubstitutional boron such as C_2_BO and CBO_2_ was also considered. Three- to five-layered graphene was synthesised using this method at 950 °C directly on copper substrates in the presence of boron powder as the boron source and ethanol as the carbon precursor with a boron content of 0.5 at.% [86]. B-GNS has also been synthesised using a copper foil substrate and triethylborane as a boron source and hexane as a carbon source at 1000 °C [87]. The XPS studies revealed that the atoms of boron were successfully incorporated into the graphitic lattice and the substitutional boron content was 1.75 at.%. Gebhardt et al. used Ni as a growth substrate, triethylborane as a boron source, and carbon tetrachloride as a carbon source [99]. The Raman analysis predicted the shift of bands to lower XPS binding energies, which were further dependent on the dopant concentration. A 1.2 eV shift was seen for a doping level of 0.3 at.%. The DFT calculations showed that atoms of boron in the framework of graphene were attached on the fcc-hollow sites of the substrate. Phenylboronic acid was used as a sole precursor by Wang et al. for the synthesis of monolayers of B-GNS, without compromising the conductivity and transmittance of the graphene films. They possessed 800 cm^2^ V^−1^ s^−1^ carrier mobility at room temperature [88]. Boric acid and polystyrene were also used as the boron and carbon sources, respectively, for the synthesis of B-GNS under the protection of H_2_/Ar with a 4.3 at.% of B [100]. Similar precursors were used by Li et al. to synthesise B-GNS and these authors observed two different bonding configurations. A graphitic carbon (BC_3_) bonding configuration where carbon atoms were replaced by boron atoms was common, but a boron silane (BC_4_) configuration occurred due to excess edge sites and defects [101].

The CVD method has an advantage that large area B-GNS are allowed to be synthesised easily [89]. However, a lack of homogeneity was the major hurdle of this method for the synthesis of single-layer doped graphene. Further, the catalysts used in the growth process, i.e., copper and nickel foil, were difficult to remove and this led to difficulties in studying the intrinsic features of doped GNS [90].

##### Self-Assembly Method

Self-assembly is a bottom-up approach where the components of an unorganised system assemble themselves to form a larger functional unit by non-covalent interactions. Selva and coworkers were able to dope reduced GO up to 10 and 20 at.% boron using B_2_O_3_ as the boron source [91]. The atomic force microscopy (AFM) images of B-GNS showed that with an increase in boron content, the surface roughness increased. The Raman spectra showed that the undoped reduced GO is n-type material whereas doped reduced GO is p-type material due to the three valence electrons of boron. Wurtz-type reductive coupling is also another example of the self-assembly approach. The reaction between carbon tetrachloride (CCl_4_) and boron tribromide (BBr_3_) in the presence of potassium resulted in the generation of B-GNS at 150–210 °C in 10 min. The amount of boron was about 2.56 at.% as determined by XPS analysis [92]. This method does not need any catalyst made up of transition metals; moreover, the extent of the dopants insertion can be easily controlled by varying the dopant amount. However, a high oxygen content was unavoidably introduced. This method offered a cost-effective means to prepare high-quality pure or doped graphene for mass production.

#### 2.2.2. Post-Treatment Methods

##### Arc-Discharge Method

An electrical breakdown of an inert gas causes an arc-discharge, which results in a prolonged electrical discharge. Panchakarla used this method in two different ways to synthesise B-GNS. In the first approach, B-GNS were synthesised employing the arc-discharge of graphite electrodes in the flow of helium (He), diborane (B_2_H_6_), and H_2_. Vapours of B_2_H_6_ were produced by the introduction of boron triflouride-diethyl etherate to sodium borohydride in the presence of tetraglyme. The B_2_H_6_ vapours were carried to the arc chamber by first passing the H_2_ at 200 Torr and then He (500 Torr) through the B_2_H_6_ generator. In the second approach, boron-packed graphite electrodes were used to conduct the arc discharge for the synthesis of B-GNS in the presence of H_2_ (200 Torr) and He (500 Torr). The XPS analysis showed that B-GNS synthesised using the second approach contained an increased amount of boron (3.1 at.%) as compared to the first approach (1.2 at.%) [102]. Palnitkar et al. also used this method to fabricate B-GNS. The XPS measurements showed that 1.2 at.% of boron was successfully doped into the graphitic lattice [93]. The preservation of graphene crystallinity and mass production are the major advantages of this method, whereas the formation of multilayer graphene, low doping level, and high current/voltage requirement are the disadvantages of this method.

##### Thermal Annealing Method

Thermal annealing is a process of altering the surface morphology of materials and the key factor determining the bonding configuration is temperature. Yeom et al. synthesised thermally reduced boron-doped graphene oxide (BT-rGO) in a two-step reaction. Firstly, a boron precursor (B_2_O_3_) was ultrasonically mixed with an aqueous solution of GO; and in the second step, thermal annealing of the mixture was carried out at four different temperatures (300, 500, 700, and 1000 °C) for simultaneous reduction and doping. The schematic representation is shown in Figure 4. The XPS analysis revealed that the peaks at 190.8 and 192.6 eV were ascribed to boron atoms of BC_3_ and BC_2_O, respectively. At 1000 °C, the peak at 190.8 eV became more predominant because of the reduction of oxygeneous moieties and the formation of more B–C bonds. High temperature favoured the formation of the B–C bonds rather than the B–O bonds. The maximum boron concentration in BT-rGO was found to be (6.04 ± 1.44) at.% at 1000 °C [94].

Wang et al. also employed this method for the fabrication of synthesised B-GNS at different temperatures and a maximum boron doping of 590 ppm was obtained at 1000 °C. These results clearly showed that with an increase in temperature, boron doping was increased [103]. Wu et al. synthesised B-GNS at 800 °C using a gaseous mixture of BCl_3_ and Ar (1:4 *v*/*v*) with a 250 mL min^−1^ of total flow rate for 2 h. The XPS peaks were obtained corresponding to BC_3_ and BC_2_O nanodomains (Figure 5). The amount of boron was about 0.88 at.%. The atomic percentage of oxygen decreased from 8.55% to 6.06% on boron doping of GNS, confirming that some oxygeneous moieties were reduced during doping. This was due to the creation of the C–B bonds [104]. A similar bonding configuration was reported by Zuo et al. using boric acid as the boron source [105].

Thermal treatment of graphite containing boron compound at 250 °C resulted in the formation of B-GNS. According to XPS studies, a single graphite crystal had ~0.22 at.% substitutional boron atoms. The insertion of substitutional boron atoms was probed by the presence of defect-induced bands in the Raman spectra. The boron atoms were 4.76 nm apart in the graphene layer [39]. Sheng et al. also synthesised B-GNS at 1200 °C using boron oxide as a boron source [90]. The AFM and high-resolution transmission electron microscopy (HR-TEM) micrographs revealed that the B-GNS was a few-layered nanostructure with a thickness of 2 nm and interplanar spacing of 0.37 nm, respectively [94]. The advantage of this method is that controllable doping can be achieved and is useful to recover the sp^2^ carbon network [106].

##### Wet Chemical and Vacuum Activation Methods

The wet chemical method is commonly called solvothermal and hydrothermal (when water is used as a solvent) and is useful for growing single crystals from a solution in an autoclave at a high temperature and pressure. Thirumal et al. used a two-step thermal approach to synthesise B-GNS. Firstly, GO was thermally reduced at 500 °C (T-GNS), and in the second step, T-GNS was treated with boric acid in a Teflon- lined autoclave to produce boron-doped graphene (HB-GNS) [107].

B-GNS was also obtained via a one-step hydrothermal method using boric acid (H_3_BO_3_) as the boron source [49]. Above 80 °C, boric acid was dehydrated and converted into boron oxide (B_2_O_3_). The boron oxide then diffused into the lattice of graphene and the carbon atoms were substituted by boron as follows [95,108,109]:(4)$$H_{3}{\mathrm{BO}}_{3}\;\to B_{2}O_{3}+\;C\;\left(\mathrm{GO}\right)\;\to {\mathrm{BC}}_{3}+{\;C}_{2}\mathrm{BO}/{\mathrm{CBO}}_{2}+{\;H}_{2}O$$

Figure 6a represents field-emission scanning electron microscopy (FE-SEM) of undoped GNS. With an increase in boron content from 0.2 to 1.0, more dense interconnected channels were observed (Figure 6b–f). The porous morphology prohibited the aggregation and restacking of individual graphene layers. It also promoted rapid electron and ion migration in 3D nanosheets.

Haque and coworkers synthesised B-GNS using benzene-1,4-diboronic acid as a boron precursor. The Raman spectrum of the boron-doped graphene organic framework (GOF) showed two bands. The D band peak position at 1351 cm^−1^ remained unaltered after boron doping, but the G band peak position was shifted to 1594 cm^−1^ from 1584 cm^−1^. The boron present in benzene-1,4-diboronic acid is electron-deficient Lewis acid, which undergoes an esterification reaction with the hydroxyl groups of GO. Thus, boron serves as an electron acceptor from GO sheets, which causes an up-shift in the G band of GOF [110]. This method is of low cost and does not involve any harsh conditions.

Few reports are available in which the vacuum activation method is used for the synthesis of boron-doped graphene nanoribbons. Xing et al. used boric acid as a boron-containing precursor to synthesise boron-doped graphene nanoribbons. Graphene maintained its decreased micrometer-sized structure after vacuum reduction and ultrasonic processing, but after doping with boron, the large-scaled graphene sheets were cut into nanoribbons. The substitutional boron doping caused a macro-residual stress, which promoted the growth of nanoribbons. The size of the graphene sheets was successfully reduced with boron doping and the conductivity of B-GNS was improved [96].

## 3. Energy Applications

### 3.1. Applications of P-GNS

P-GNS has been studied for its applications in electronic devices [79,82], energy storage devices [77], and electrocatalysis [75,80], etc., due to the difference in the properties of the phosphorous and carbon atoms. These applications are generally divided into four categories, viz. lithium-ion batteries, solar cells, supercapacitors, and fuel cells. The applications of P-GNS synthesised using different approaches are listed in Table 1. In the next subsection, each application is elaborated.

#### 3.1.1. Fuel Cells

Fuel cells are used to convert the energy from a chemical nature to an electrical nature. Hydrogen gas is fed into the anodic compartment where it is oxidised and the oxygen gas is fed into the cathodic compartment where it is reduced. The ORR is the rate-determining step that occurs in the cathodic compartment [111]. ORR can occur in the basic medium via two different routes: (i) either by a four-electron mechanism, or (ii) by two-electron pathway. Each electrode is made up of porous compressed carbon containing a small amount of catalyst to increase the rate of ORR. The most popular catalyst for the ORR process is platinum; however, the high cost prevents the use of platinum in fuel cells on a commercial scale. Pristine graphene is not effective towards ORR because it is inefficient to enable electron transfer. Because of the charge-polarization of the binding between the dopant and carbon, heteroatom dopants speed up the adsorption of oxygen on the cathode and the breaking of the oxygen–oxygen bond (Figure 7).

The poisons such as carbon monoxide (CO) and methanol (CH_3_OH) in electrolyte inhibits the ORR at the platinum surface. This can be avoided by metal-free electrocatalyst such as P-GNS. Phosphorous atoms increase the absorption and cleavage of the oxygen– oxygen bond due to the charge polarization of the heteroatom and carbon bond. Qiao and coworkers reported that dual-doped graphene (PN-GNS) synthesised by pyrolysis of GO (as the carbon source) and diammonium hydrogen phosphate (as the nitrogen and phosphorous source) acted as a better electrocatalyst as compared to single-doped and undoped graphene [112]. The synergistic effect, fast electron transfer by highly conducting graphene, and the 3D porous structure provided porous channels that made the inner active sites accessible to the oxygen molecules and the electrolyte. Similar results were observed by Jo’s group during the use of BP-GNS as an electrocatalyst in the fuel cells [113].

The ORR activity of dual-doped graphene can be further improved by their fabrication with inexpensive perovskite such as (La_0_._8_Sr_0_._2_MnO_3_). The introduction of perovskite to undoped graphene increased the number of electrons (n) moved from anode to cathode electrode 2.7 to 3.6 at −0.5 V, but it did not alter the current density. This problem was overcome by fabricating perovskite with dual-doped graphene. Here the n value increased to 3.8 at −0.5 V. This may be due to acceleration or conductivity effects provided by perovskite. The kinetic analysis showed that the direct four-electron pathway was favoured [114].

With the introduction of phosphorous into co-doped graphene NS-GNS, a large increase in ORR performance was observed because of the formation of a conductive P–N bond. The ternary-doped graphene (PNS-GNS) showed excellent electrocatalytic activity, which was five to six times higher than P-GNS and two times higher than NS-GNS [115]. A similar method was used by Zhang et al. to use PNF-GNS as an electrocatalyst for fuel cells. Fluorine, a highly electronegative element, is a promising dopant to produce charge polarization that can significantly enhance the electrocatalytic activity for graphitic carbon atoms [116]. The heteroatom carbon species synergistically affects the performance of dual- and ternary-doped graphene.

Nitrogen-, iron-, sulphur-, and phosphorous-doped graphene was synthesised using the direct pyrolysis of *Shewanella oneidensis* bacteria, displaying a comparable catalytic current density to Pt/C [117]. The presence of metals in an electrocatalyst has been associated with several difficulties, viz. low selectivity, poor stability, a negative environmental impact, and higher cost as compared to metal-free catalysts [118]. The enhanced ORR activity possessed by the quaternary-doped graphene (BPNS -GNS) vs. the mono-doped graphene was due to two reasons: firstly, the simultaneous insertion of various heteroatoms into the lattice resulted in a synergistic effect; secondly, the ORR activity was strongly influenced by the formation of bonds between different heteroatoms. The P–N moieties formation increased the number of active locations for the adsorption of oxygen gas molecules as compared to the B–N species [117]. The BPNS-GNS showed outstanding current density values, number of electrons transferred during ORR, and durability as compared to dual- and ternary-doped graphene.

#### 3.1.2. Lithium-Metal Batteries

Lithium-ion batteries are important energy-producing and storage devices. These are rechargeable batteries, which contain lithium solution as an electrolyte. The importance of lithium-ion batteries in energy-related devices nowadays can be understood from the Nobel Prize given in 2019 to three scientists, Stanley Whittingham, John Goodenough, and Akira Yoshino, for their innovation in the synthesis of lithium-ion batteries. The use of these lightweight, rechargeable, and powerful lithium-ion batteries from electronic gadgets to electric vehicles have revolutionized our lives. The negative electrode of a conventional lithium-ion cell is composed of carbon-based materials, while the positive electrode is a metal oxide. During charging, ions move from anode to cathode and the reverse happens during discharging. Due to its low affinity for lithium atoms and the tendency of deposited lithium atoms to cluster on the graphene surface, pure graphene is not a good choice for lithium storage [106]. Moreover, the penetration of lithium ions in GNS requires high energy, which can be provided by employing modified graphene and carbon nanomaterials as electrodes. Lithium ions can penetrate graphene materials and lithium clustering is prevented by the presence of defects. Since the lithium atom donates electrons, graphene can be doped with an electron-poor element like boron to boost its storage capacity. However, due to the increased binding energy between lithium and B-GNS, it restricts delithiation (lithium diffusion). In summary, these materials boost battery capacity; however, delithiation is ineffective. Although the storage capacity of graphene is reduced as a result of doping it with electron-rich atoms like nitrogen, phosphorus, and metal oxides, the delithiation process is more effective. This can be caused by reduced electrostatic repulsion between lithium and dopants/ metal oxides and their binding energy. Therefore, the charge/discharge performance is increased by these materials (Figure 8).

Zhang’s group used P-GNS in lithium-ion batteries as an anode material for the first time. They also compared the results with undoped graphene and found that P-GNS acted as a superb electrode because of various topological defects introduced on its surface during the doping process. This led to a randomized carbon lattice that improved the lithium insertion characteristics. P-GNS has a considerably larger reversible discharge and charge capacity than undoped graphene (approximately 280 mAh g^−1^), with nearly no loss across 80 cycles, indicating that P-GNS-based electrodes possess a good longevity [119]. Luan et al. [120] showed that PN-GNS has a high reversible charge and discharge capacity along with an excellent cycling rate after 600 cycles. The specific capacity of PN-GNS was 889 mAh g^−1^ at a current density of 1000 mA g^−1^. Due to low abundance and uneven global distribution of lithium ions, the increasing demand of Li-ion batteries can hardly be meet.

#### 3.1.3. Alkaline Ion Batteries

P-GNS is also a viable option for anodes in sodium-ion batteries [121]. Na-ion batteries have similar characteristics as Li-ion batteries. Due to its distinct ultrathin and wrinkled shape, P-GNS as an anode material demonstrated superior cycle stability and excellent rate performance. At a current density of 50 mA g^−1^, the specific capacity of P-GNS was calculated to be 284.8 mAh g^−1^. Due to synergistic effects such as increased interlayer distance, improved electrical conductivity, heteroatomic defects, and good electrode/electrolyte wettability, P-GNS maintained its specific capacity after 600 cycles. A high specific capacity and extremely high cycling stability are both characteristics of graphene with phosphorus and oxygen dual-doping (PO-GNS) (474 mA h g^−1^ after 50 cycles at 50 mA g^−1^ and 160 mA h g^−1^ after 600 cycles at 2000 mA g^−1^). This might occur as a result of the dopants’ high inter-layer spacing, which aided the insertion and extraction of potassium ions [122].

#### 3.1.4. Supercapacitors

Supercapacitors have high capacitance values as compared to conventional capacitors because they link the gap between rechargeable batteries and electrolytic capacitors. They are also known as gold caps, ultracapacitors, or super caps. Supercapacitors come in a variety of forms, including hybrid capacitors, electrochemical pseudocapacitors, and electrostatic double-layer capacitors. The first type of supercapacitors use carbon materials for electrode preparation, and charge separation is achieved using the Helmholtz double layer at the junction of the electrolyte and the surface of a conductive electrode. In electrochemical pseudocapacitors, conducting polymers and metal oxide electrodes are used and charge separation is achieved by Faradaic electron charge transfer with electrosorption, intercalation, and redox reactions. In the third type, hybrid capacitors are a combination of both electrodes; one that stores the charge faradaically and the other stores the charge electrostatically.

Due to graphene’s enormous surface area and large electric double-layer capacitance, it has recently been employed as an electrode in supercapacitors [123]. However, pristine graphene is chemically inert, which encounters some challenges as it does not give electrochemical capacitance, i.e., pseudocapacitance [106]. For usage in supercapacitors, doping can introduce some significant amounts of deviations in the graphene lattice. Heteroatom-doped graphene was more advantageous when used in supercapacitors as it enhanced conductivity, improved stability, and had good reactivity compared to pristine graphene.

Graphene doped with phosphorous exhibited improved capacitive performance in aqueous electrolyte due to higher pseudocapacitance and a strong conducting network in the graphene structure. A supercapacitor was created by Karthika et al. [124], utilising P-GNS, and it displayed a high power density of 9 kW kg^−1^, a high specific capacitance of 367 Fg^−1^, and a high energy density of 59 Wh kg^−1^. After many hundreds of cycles, there was no discernible loss in any particular capacitance. Thirumal et al. [82] also synthesised P-GNS using the electrochemical method, where the specific capacitance was 290 Fg^−1^ at a current density of 0.5 Ag^−1^. Thus, the different approaches affect the capacitance values because different defects are created by the different methods. DFT relates the enhanced specific capacitance of P-GNS with structural changes in the lattice. DFT calculations were used by Song et al. [79] to identify two distinct mechanisms: first, the rise in interlayer separation following doping; and second, the high polarity of the P–O groups. Electrolyte filled the gap between the graphene sheets as their distance from one another grew, forming additional double layers that may store charge.

Polyaniline–graphene-based electrodes exhibited high specific capacitance of 480 Fg^−1^ at a current density of 1 Ag^−1^ [125]. This value can be further enhanced by introducing heteroatoms. Nanocomposite of polyaniline with P-GNS showed high specific capacitance (603 Fg^−1^ at 1 Ag^−1^), which was six times higher than bare polyaniline. Moreover, this value further increased with an increase in current density from 1 to 15 Ag^−1^ [126].

Because of the synergistic effects of the heteroatom dopants, co-doping of heteroatoms has been seen as a potential way to improve the energy storage performance of graphene-based materials. Excellent specific capacitance of 183 Fg^−1^ at 0.05 Ag^−1^ current density was present in PN-GNS. P-GNS is a superior supercapacitor electrode than a broad potential window of 1.6 V in an aqueous electrolyte because it contains functional groups that include phosphorus [127].

### 3.2. Applications of B-GNS

The applications of B-GNS are discussed in different subsections, viz. solar cells, sensors, supercapacitors, lithium-ion batteries, fuel cells, and photocatalysts. The properties of the materials are briefly discussed in next subsections.

#### 3.2.1. Fuel Cells

The substituted boron atoms in the graphene lattice acted as active sites for adsorption of oxygen and speeding up oxygen–oxygen bond breaking. Molina-Garcia et al. [114] observed that binary-doped graphene improved conductivity, making it a suitable catalyst for ORR activity. This might be the result of flaws that developed as a result of the doped graphene, which increased the electrical conductivity of the perovskite materials in the end. The kinetic results depicted that the four-electron pathway was followed. As an alternative to Pt-based materials with high costs, multiple doping of graphene can be used to create an electrocatalyst that is affordable, dependable, and metal free. Co-doping graphene with various atoms may help it perform even better electrocatalytically in the ORR process.

Wu et al. [128] carried out a structural modification of graphene with boron and nitrogen dopants using the hydrothermal self-assembly method. The experimental results revealed that edge-rich BN-GNS without an inert covalent B–N bond showed more superior ORR electrocatalytic performance as compared to BN-GNS with fewer edges and an inert B–N covalent bond [128]. Qin et al. [83] prepared BN-GNS to study the adsorption and reduction of oxygen on a cathode using DFT studies. Their calculations indicated that the BC_3_ and graphitic N were the main active sites among various boron doping or/and nitrogen configurations. Mazanek et al. [1] also synthesised BN-GNS using thermal exfoliation of GO and studied its electrocatalytic performance for ORR process. They determined the atomic percentage of boron and nitrogen using ICAP-AES analysis. The boron content (4.33 to 6.53%) was higher than the nitrogen content (0.11 to 0.53%). They found that ORR performance was related to intrinsic doping as well as metallic impurities introduced during the synthesis of GO, such as manganese, chlorate, permanganate, etc. In order to use BN-GNS as a metal-free electrocatalyst in fuel cells, Il and colleagues synthesised BN-GNS. The electron transfer number (n) in the 0.225–0.465 V potential range was 3.53–3.84, indicating that the BN–GNS preferred the four-electron pathway for oxygen reduction. These findings showed that doped graphene had the potential to take the place of pricey precious metal catalysts [129].

With a high current density (55 A cm^−1^) and a low onset potential (−0.10 V), BNP-GNS demonstrated excellent electrocatalytic activity [130]. Due to the incorporation of various dopants with various electronegativities, quaternary-doped graphene (BPNS-GNS) demonstrated superior ORR catalytic activity in comparison to single- and dual-doped graphenes. They were able to change the oxygen molecule’s binding energy, which made oxygen dissociation easier [117].

#### 3.2.2. Solar Cells

The solar cell is an electrical device that changes sunlight or artificial light into electrical energy through the photovoltaic effect and thus it is also known as the photovoltaic cell. The absorption of light energy, which produces excitons or electron–hole pairs, is one of the three fundamental characteristics needed for a solar cell to function. The separation of charge carriers of different types is the second step, and the independent extraction of those carriers to an external circuit is the third step. Dye-sensitized, organic, quantum dot, perovskite, and perovskite solar cells are examples of common solar cells. The four layers of a solar cell are the interface layer (Au, indium tin oxide), hole transport layer (2,2′,7,7′-tetrakis (N,N-di-4-methoxy-phenylamino)-9,9′,-spiro bi fluorene), electron transport layer (TiO_2_), and transparent conducting layer (fluorine tin oxide) [91]. The power conversion efficiency (PCE), which is determined by the percentage of incident power converted into electricity, is used to calculate the total efficiency of solar cells. In solar cells with PCE of 7.09–8.96%, p-type material like B-rGO serves as a hole transport layer [91]. A B-GNS/SiO_2_-based solar cell with an open current voltage of 0.53 V, and a short current density of 18.8 mA cm^−2^ was developed by Li et al. [101] under one solar irradiation. A nitric acid treatment of B-GNS considerably improved the performance of the solar cell. Nitric ions added more P-doping, improving the electrical conductivity, and decreasing the charge transfer resistance [101,131]. These studies imply that doped graphene can function as an efficient hole transport material in the active layer of solar cells.

#### 3.2.3. Lithium-Metal Batteries

Due to their low memory loss, high energy density, and low self-discharge rate, rechargeable lithium-ion batteries are extensively employed in portable electronic devices [132]. Using a hydrogen-assisted reduction process, Sahoo et al. created SnO_2_-adorned B-GNS for use as a lithium-ion battery anode material. In comparison to bare SnO_2_ and B-GNS, they discovered that the synthesised hybrid had the largest reversible capacity of 744 mAh g^−1^ at a current density of 100 mAh g^−1^. An improved lithium interaction resulted from the SnO_2_, a spacer within the GNS, increasing the likelihood of lithium intercalation and alloying against re-stacking of the 2D planes. Due to the wrinkled graphene network and homogenous dispersion of tiny SnO_2_ nanoparticles (NPs), a rise in reversible capacity was seen with increasing boron content in B-GNS [23].

Silver (Ag) NPs embedded on B-rGO served as the anode material in a nanocomposite that was synthesised by Bindumandhavan et al. [64]. Due to the interaction between B-GNS and Ag NPs, this type of anode material demonstrated a substantial reversible capacity (1484 mAh g^−1^ at 50 mA g^−1^). This anode showed increased cycle stability and reversible capacity of 540 mAh g^−1^ at 100 mAh g^−1^ after 100 consecutive cycles. The presence of defect sites and heteroatoms in the presence of boron and Ag NPs was attributed to these outstanding electrochemical performances. Tian et al. [133] have also explored the possibilities of boron- and sulphur-doped reduced graphene oxide (BS-rGO) as a cathode in lithium-ion batteries. According to their findings, a hydrothermally produced BS-rGO cathode outperformed S-rGO in terms of reversible discharge capacity, achieving 521 mAh g^−1^ at 0.1 C after 100 cycles. The excellent high rate of performance was attributed to the B-strong rGO’s electrical conductivity. The composite cathode’s electric conductivity was enhanced and the shuttle effect was delayed by boron doping on rGO. Due to the co-dopants’ synergistic effects, BN-GNS produced by the hydrothermal process performed better than single-doped graphene. For the objective of preserving active material during the cycle and enhancing the carbon’s wettability in the organic electrolyte, borax and nitrogen dopants had significant binding opportunities with polysulfide species [134]. The next-generation energy storage technology for lithium-sulfur batteries is thought to be promising [135].

#### 3.2.4. Alkaline Ion Batteries

Potassium-ion batteries are gaining popularity as lithium-ion battery substitutes due to the abundant availability of potassium in the Earth’s crust. B-GNS has the potential to be used as an anode material for potassium-ion batteries due to its high specific capacity of 546 mAh g^−1^ and low migration barrier of 0.07 eV. This may be due to the substrate being doped with boron, which causes it to become electron-deficient and causes a sizable charge transfer from potassium to the substrate. Due to the inhibition of dendritic development and prevention of potassium atom clustering, there is good cycling stability [136,137].

#### 3.2.5. Supercapacitors

Supercapacitors based on doped graphene electrodes showed higher interfacial capacitance as compared to undoped ones. Some researchers reported that the presence of dopant on the graphitic lattice might lead to electrochemical reduction–oxidation reactions on the surface, which would ultimately enhance the capacitance of the material. The specific capacitance of boron-doped reduced GO was 448 Fg^−1^, which was three times higher than reduced GO (135 Fg^−1^). A supercapacitor’s specific capacitance typically depends on the electrode’s BET surface area and electrical conductivity, both of which decrease as the annealing temperature rises. On the other hand, boron-doped GO lowered the electrical conductivity and BET surface area increased as the annealing temperature rose from 300 to 1000 °C. At a higher temperature (1000 °C), the doping of boron heteroatoms may create defect-like small pores in the graphene nanosheet’s plane that prevents the growth of graphitic structure [94].

Thirumal et al. performed galvanometric charge–discharge and cyclic voltammetry (CV) measurements on thermally reduced graphene and B-GNS in 0.5 M H_2_SO_4_ at various current densities (1 Ag^−1^ to 4 Ag^−1^). The curves of charge–discharge were triangular in shape (Figure 9a), and CV curves were rectangular in shape (Figure 9b), which suggested that both electrodes showed double-layer capacitance behaviour. The highest specific capacitance for thermally reduced graphene and B-GNS was 52 and 113 Fg^−1^, respectively (Figure 9c,d), which showed that B-GNS can act as a better electrode as compared to undoped GNS for supercapacitor applications [95]. However, Mombeshora et al. reported contradicting results [138]. This research group prepared B-GNS using the thermal reduction method in the presence of sodium borohydride [138]. During the reduction process, defects were created that acted as charge-trapping sites.

Numerous publications also examined the effects of co-doping graphene with two or more heteroatoms. Chen et al. produced BN-GNS with atoms ranging from 0.6% to 2.1% (B) and 1.74 to 2.56% (N). In order to get a satisfactory performance in electrochemical energy storage, dopants were simultaneously introduced, creating a synergistic effect between boron and nitrogen [97]. The findings indicated that while boron doping enhances the performance of supercapacitors at high current densities, nitrogen doping can guarantee high capacitance.

## 4. Conclusions and Future Outlook

Heteroatom doping is one of the best techniques for altering the properties of graphene. Depending on the type of heteroatom dopant such as boron, nitrogen, phosphorous, sulphur, and fluorine, and doping configurations, heteroatom-doped graphene nanomaterials can be used for a spectrum of applications.

In this review, we have discussed direct and post-treatment methods for the synthesis of P-GNS and B-GNS. As compared to B-GNS, research on P-GNS is still in its infancy and the comparative evaluation of heteroatom doping on the structural, textural, and morphological properties is still unexplained. Various characterisation tools, viz. XPS, TEM, SEM, Raman coupled with DFT calculations, have been used to study the structural properties of doped graphene-based nanomaterials. The starting dopant material, precursors, temperature, reaction time, and other factors all affect the characteristics of the obtained materials. However, due to the heterogeneity of the doped graphene, conflicting results are reported and the current understanding of the characteristics of heteroatom-doped graphene is still insufficient. The innovative uses of doped graphene materials for energy conversion, energy storage, and gas storage have been addressed. Because of their fast electron transport and three-dimensional porous structure, materials based on doped graphene perform better in energy-related devices. Because of the additional possibilities offered by heteroatom doping, graphene research will continue to flourish. Despite the positive advancements made in the field of doped graphene-based materials, there are still numerous obstacles that need to be overcome, viz. the limited number of publications demonstrating their applications towards the environmental field and biological toxicity. These issues need to be critically considered to extend their practical applications and for their large-scale applications.

## Data Availability

Not applicable.

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
