# Peer review of "Phosphorous- and Boron-Doped Graphene-Based Nanomaterials for Energy-Related Applications"

_materials, 2023, doi:10.3390/ma16031155_

Round 1

Reviewer 1 Report

The review paper discusses about the boron and phosphorous doping to graphene and their application in fuels cell, batteries, and supercapacitors. This is an important aspect of to control graphene properties for various applications. Since the review focus on only doping of P and B atoms in to the graphene, so Electrochemical intercalation and exfoliation process to be included. As it is a green methodology for doping graphene. So authors are requested to insert a section on electrochemical way of doping in the manuscript.

Author Response

Respected reviewer

Thanks for the comments and suggestions

The point wise replies are attached in the word file

Best regards

Reviewer 2 Report

Although your review is instructive about graphene doping, graphene, graphene-oxide and reduced graphene-oxide are sometimes used interchangeably, which causes confusion in the manuscript. In addition, there are some sloppiness in the figures and in the text. The deficiencies that I have identified are presented in the file.

In addition, without the bibliography, the similarity is as high as 24%. It is a high value.

Author Response

(The authors gave the same response as above.)

Round 2

Reviewer 1 Report

It can be published now in its present form.

Reviewer 2 Report

The corrections you made as a result of the evaluations of the reviewers were found positive and appropriate.